

# Evaluation of a New Observationally Based Channel Parameterization for the National Water Model

Aaron Heldmyer[1], Ben Livneh[1,2], James McCreight[3], Laura Read[3], Joseph Kasprzyk[1], and Toby Minear[2]

[1]Department of Civil, Environmental, and Architectural Engineering, University of Colorado Boulder, Boulder, 80309, USA
[2]Cooperative Institute for Research in Environmental Sciences, University of Colorado Boulder, Boulder, 80309, USA
[3]National Center for Atmospheric Research, Boulder, 80305, USA

*Correspondence to*: Aaron Heldmyer (aheldmyer@usgs.gov)



**Abstract.** Accurate representation of channel properties is important for forecasting in hydrologic models, as it affects height, celerity, and attenuation of flood waves. Yet, considerable uncertainty in the parameterization of channel geometry and hydraulic roughness (Manning's n) exists within the NOAA National Water Model (NWM), due largely to data scarcity: only ~2,800 out of the 2.7 million river reach segments in the NWM have measured channel properties. In this study, we seek to improve channel representativeness by updating channel geometry and roughness parameters using a large, previously
unpublished hydraulic geometry (HyG) dataset of approximately 48,000 gages. We begin with a Sobol' sensitivity analysis of channel geometry parameters for 12 small semi-natural basins across the continental U.S., which reveals an outsized sensitivity of simulated flow to Manning's n relative to channel geometry parameters. We then develop and evaluate a set of regression-based regionalizations of channel parameters estimated using the HyG dataset. Finally, we compare the model output generated from updated channel parameter sets to observations and the current NWM v2.1 parameterization. We find that, while the
NWM land surface model holds the most influence over flow given its control over total volume, the updated channel parameterization leads to improvements in simulated streamflow performance relative to observed flows, with a statistically significant mean $R^2$ increase from 0.479 to 0.494 across approximately 7,400 gage locations. HyG-based channel geometry and roughness provide a substantial overall improvement in channel representation over the default parameterization, updating the previous set value for most reaches of Manning's n = 0.060 to a new range between 0.006 and 0.537 (median 0.077). This
research provides a more representative, observationally based channel parameter dataset for the NWM routing module, as well as new insight into the influence of the routing module within the overall modeling framework.

## 1 Introduction

In the continental United States (CONUS), flood events are among the most significant natural disasters in terms of damage
to life and property. Direct losses from flooding rank a close second to hurricanes and represent a quarter of nationwide total damages stemming from natural hazards at $144 billion in losses from 1960 to 2009 (Gall et al., 2011). Flood waves generated from extreme precipitation events or infrastructure failure propagate from the origin along a channel network and are influenced by the geometric and physical properties of the channels along its path. Forecast centers simulate hydrologic processes using a framework of atmospheric and hydrologic models coupled with routing models to simulate flood wave
propagation, and parameterization of channel properties within these models is necessary for forecasting of flood waves and thus mitigation of potential damage. Sparse observational data availability renders the adequate characterization of channel properties a challenging task and typically requires some form of parameter regionalization. In this study, we seek to improve flood simulation accuracy of the NWM by replacing its current channel parameters with those based on a regionalization of an extensive observational database. This research is focused on the NWM channel routing module, and therefore does not
investigate parameterization of the LSM, the gridded routing module, or any other component of the NWM framework.





Agencies such as the National Oceanic and Atmospheric Administration (NOAA) supply much of the actionable flood forecasting data for informed policymaking and emergency management decisions. In many cases, these data are produced by land surface models (LSMs) continuously forced by weather forecast data. This framework allows for the simulation of hydrologic processes occurring at individual watersheds forecast into the near future to produce actionable, time-sensitive

hydrologic information. One emerging hydrologic modeling framework is the NOAA National Water Model (NWM). Launched in 2016, the NWM continuously simulates observed and forecast streamflow for approximately 2.7 million river reaches over CONUS. The basis of the NWM is the Weather Research and Forecasting Hydrologic Model (WRF-Hydro; Gochis et al., 2020), which accepts forcing data from a number of different sources to generate short- (18 hour), medium- (~10 day), and long-range (30 day) forecasts, as well as analysis of current streamflow. WRF-Hydro is one-way or two-way coupled

(depending on configuration) with the Noah Multi-Parameterization (Noah-MP; Niu et al., 2011) LSM to simulate land surface processes at 1 km resolution, and separate two-part channel routing system . The first part routes flow on a 250 m grid using both diffusive wave surface and saturated subsurface flow routing. The second routes flow along the National Hydrography Dataset (NHD)-Plus medium resolution channel network using the Muskingum-Cunge method (Cunge, 1969) of flow routing. The two-part routing system (gridded and NHD network-based) employed by the NWM represents a higher degree of

sophistication compared to most other mainstream operational models. For example, the Sacramento Soil Moisture Accounting Model (SAC-SMA; Burnash et al., 1973) does not implicitly route flow between conceptual reservoirs, and the Hydrology Laboratory's Research Distributed Hydrologic Model (HL-RDHM; Koren et al., 2004) assumes uniform, conceptual hillslopes within a relatively coarse 4 km x 4 km grid within its hillslope and channel routing module (Fares et al., 2014). Additionally, the channel routing component in HL-RDHM relies on a unique relationship between discharge and cross-sectional area for

each cell dependent on just four parameters (slope, a roughness coefficient, a shape parameter, and a top width parameter). To contrast, channels within the NWM NHD network-based routing module are conceptualized using a trapezoidal geometry described by 11 parameters such as top width, bottom width, side slope, and Manning's n. These parameters are required for all 2.7 million modeled reaches across CONUS, and therefore necessitate a significant amount of data for accurate channel representation. Currently, there is likely significant uncertainty in channel parameters due to a sparsity of data available for

inferring them. Approximately 2,800 reaches containing physical measurements are used to inform routing module parameters. Additional observational data may enhance representation of the routing module, thereby improving flood forecasts. The hydraulic geometry (HyG) dataset is a new, unpublished collection of approximately 2.8 million field discharge measurements from roughly 48,000 gages well-distributed across CONUS, comprising discharge measurements from both active and inactive gages, as well as eight state-wide datasets. HyG was a result of development originating from the smaller USGS 'HydroSWOT'

database (Canova et al., 2016), a stream bathymetry and hydraulic properties database from acoustic Doppler current profiler data compiled for hydrologic modeling by the NASA Surface Water and Ocean Topography (SWOT) mission. While HyG is gage-based and thus spatially discontinuous, the HyG collection is a significant source of large-scale stream bathymetry and hydraulic data, representing a 20-fold increase in observations compared to other databases. This catalog is likely to only be





surpassed after remote sensing platforms are capable of achieving higher precisions, such as the NASA SWOT mission
(Biancamaria et al., 2016), still a year or more away.

While HyG may be a significant improvement over the current observational database used by the NWM, the utilization of
HyG across CONUS requires the estimation of channel properties where observations are not available. This form of parameter
transfer is often termed 'regionalization'. Regionalization is defined here as the transfer of parameters estimated at observed
spatial units to unobserved units under the assumption of hydrologic similarity. Largely due to the diversity of contexts in
which regionalization techniques are typically applied, there is no consensus on which technique is "best" (Ayzel et al., 2017).
A wide variety of regionalization techniques have historically been developed to make estimates at ungaged locations, though
most may be broadly categorized into one of two main forms: distance-based and regression-based (He and Wilkerson, 2011;
Livneh *et al.*, 2013).

The first group of regionalization methods is based on distance, premised on the notion that parameters are continuously
distributed through space, and similarity between two arbitrary points is correlated with spatial proximity. The spatial structure
of this correlation is modeled with varying types of interpolation, and the underlying statistical basis of these models varies
widely. Typical regionalization methods which fall under this category include the method of inverse distance weighting
(IDW), the nearest-neighbor (NN) method, and the method of Kriging, with the latter two generally considered the most widely
used (Ayzel et al., 2017). For the specific application to the NWM channel network, grid-based spatial interpolations of channel
parameters may be inapplicable. While one routing module in the NWM framework does route flow on a 250 m grid, the river
network within the routing module of interest is not represented as a spatially continuous grid, but rather a dendritic network
of features overlaid on a spatially continuous land surface. In this case, two seemingly proximal channels may instead be
distant from the perspective of the network, and consequently have dissimilar properties due to natural variations in geology
and terrain, vegetation, and development-related disturbances (e.g., urban drainage systems).

The second group of regionalization methods is not constrained by spatial proximity, and instead seeks to transfer parameters
on the basis of similarity in physiographic features (land cover, soil, slope, etc.). Rather than spatial interpolation, similar
catchment features can be found across long distances, such that the regionalization proceeds along dimensions of similar
hydrologic features rather than distance. Regression-based approaches are examples of this category, and are typically of linear
form (e.g., Gitau and Chaubey, 2010; Heuvelmans et al., 2006), though non-linear, weighted, and sequential forms have also
been applied (e.g., Abdulla and Lettenmaier, 1997; Kay et al., 2006; Li et al., 2010). Regional scale regression curves for
channel geometry, first developed by Dunne and Leopold, (1978), operate on the assumption of similarity in geology, soil,
climate, and hydrology within the region (Bieger et al., 2015). The current implementation of the NWM routing module
parameterizes channel geometry through regression-based regional curves relating channel top width with NHD-Plus drainage
area following the method of Blackburn-Lynch et al. (2017). Hydraulic roughness (Manning's n) is currently based on expert
opinion and a function of Strahler stream order. Updating these relatively simplistic regionalization approaches using new
relationships across variable spatial scales may serve to improve estimation at ungaged reaches.





In this study, we hypothesize that enhancements in simulated streamflow goodness-of-fit (GOF) metrics performance are possible through an update to the NWM channel routing geometry and hydraulic roughness parameters. Therefore, the objectives are to 1) better characterize the influence of channel parameters on NWM simulated streamflow, 2) develop a
regionalization strategy for the HyG dataset such that a spatially complete and representative parameter dataset may be developed, and 3) examine the effects of this regionalized dataset on model flow GOF metrics performance. A greatly expanded database of channel geometry and hydraulic roughness regionalized to unobserved reaches may represent a substantial improvement to channel parameters in the NWM. Furthermore, assessing the degree of improvement attainable from updating channel routing parameters addresses a knowledge gap relevant to future model calibration efforts.

## 115   2 Methods

### 2.1 Overview

The analysis begins with a description of selected NWM channel parameters and transformations where applicable (Sec 2.2). A sensitivity analysis of these channel parameters is then conducted to determine influence of channel parameters on simulated streamflow (Sec 2.3). Following this, channel parameters are developed from HyG data at observed locations (Sec 2.4) and
regionalized through multi-scale regression-based approaches to all 2.7 million reaches CONUS-wide (Sec 2.5). Finally, we update the NWM routing model with the regionalized parameter sets and evaluate differences in model performance via streamflow simulations and direct errors at gaging stations in the representative basins (Sec. 2.6).

### 2.2 Channel Parameters

Channels within the NWM routing module are represented by a compound trapezoidal geometry (Figure 1), consisting of a
main channel that carries base flow and runoff up to bank-full flow conditions, and a conceptual floodplain which carries overbank flow in times of flooding. For examination in the sensitivity analysis, six parameters from the routing module that describe the channel dimensions were selected—bottom width (BW), top width (TW), floodplain top width ($TW_{cc}$), and channel side slope (m)—along with the Manning's n roughness coefficient for both the main channel (n) and floodplain ($n_{cc}$). Within this parameter set, there is a physically based, ascending relationship between BW, TW, and $TW_{cc}$ (i.e., BW < TW <
$TW_{cc}$). This presents an issue for the sensitivity analysis, which is most effective when sampling variables independently. Therefore, two new parameters, channel depth (d) and floodplain change in width ($dx_{cc}$), were created, such that TW and $TW_{cc}$ are calculated as a function of these new parameters:

$$dx_{cc} = \frac{TW_{cc} - TW}{2} \qquad (1)$$

$$d = \frac{TW - BW}{2} \times m \qquad (2)$$

Thus, a final set of independent parameters including BW, d, $dx_{cc}$, m, n, and $n_{cc}$ was established for examination in the sensitivity analysis.



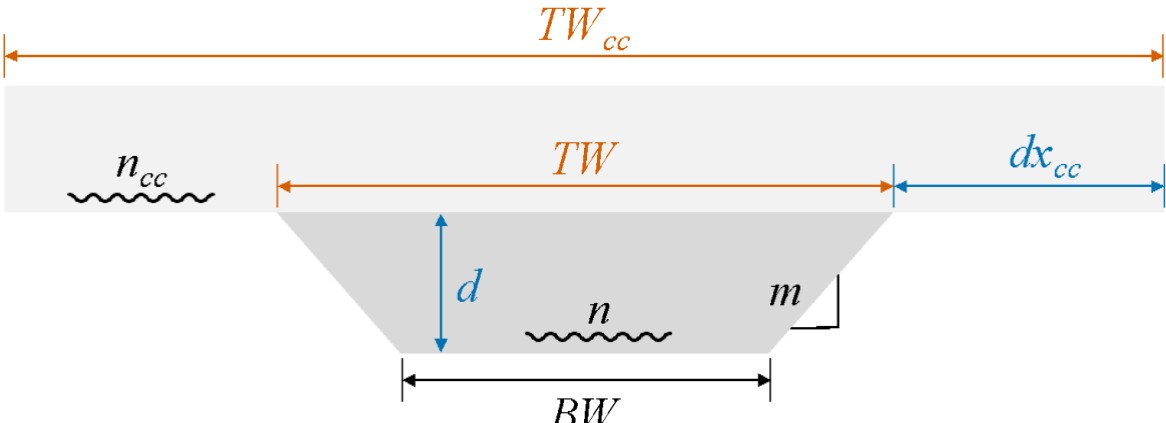

**Figure 1: Cross-sectional diagram of the trapezoidal channel schematic used in the channel routing module of the NWM. This compound channel representation consists of a main channel (dark gray) and a floodplain (light gray) which becomes inundated in times of overbank flooding in the main channel. Parameters in blue were used to compute the parameters in orange for consistency**
**among inputs for the sensitivity analysis. Parameters in black remained unchanged.**

## 2.3 Sensitivity Analysis

A sensitivity analysis was conducted to establish the influence of channel parameters on model streamflow output (Pianosi et al., 2016). To generate combinations of values within the parameter set, a Latin Hypercube Sampling (LHS) method was used
to systematically sample across a hyperdimensional space. LHS is based on the "Latin square" design, which contains a single sample in each row and column of a hypothetical square with edges representing the ranges of two parameters (McKay et al., 1979). In this method, cumulative density functions (CDFs) for each parameter are divided into equal partitions, and data points within each partition are selected and randomly combined with other selected parameter values. LHS was chosen as it offers an advantage over random sampling techniques by ensuring representativeness of the real variability among parameters
of each randomly selected combination.

Given a lack of strict boundary conditions for the parameter values, inputs were instead varied as a function of their nominal values developed from regional curve relationships with drainage area (for estimating geometry parameters) following Blackburn-Lynch et al., (2017), and expert opinion scaled by Strahler stream order (for estimating Manning's n parameters). These were compared to the resulting variation in model output expressed as a fraction of the output under the default
parameterization. Parameters were modified between a factor of 0.1 and 10 of their nominal values in an effort to encompass the range of possible error in parameter values. Uniform distributions of parameter scalars in the [0.1, 10] space were generated and combined using the 'randomLHS' function of the 'lhs' R package (Carnell, 2020) and subsequently multiplied with the





relevant default parameters. Here, d and $dx_{cc}$ were calculated from the original data using Eqns. 1-2, combined with multipliers, and transformed back to the original parameter space.

We employ the variance-based method of Sobol' (Sobol′, 2001) for analysis of the NWM channel routing module parameter sensitivity following the precedent set by many prior sensitivity analyses of hydrologic models (e.g., Abebe et al., 2010; Baroni and Tarantola, 2014; Cibin et al., 2010; Herman et al., 2013; Massmann and Holzmann, 2012; Nossent et al., 2011; Pappenberger et al., 2008; Reusser et al., 2011; Song et al., 2012; Tang et al., 2007; Wagener et al., 2009; Yang, 2011; Zelelew and Alfredsen, 2013). Specifically, we follow the method of Saltelli (2002) using the 'sobolSalt' function within the

'sensitivity' R module (Iooss et al., 2021) to estimate first order (the influence of each parameter alone) and total effect (first order plus all interactive effects) indices, which implements a Monte Carlo estimation of the Sobol' indices at a cost of $n*(p + 2)$ evaluations, where n is sample size and p is the number of parameters.

A total of n = 3,360 unique channel parameter sets (70 groups of 48 members each) for p = 6 parameters were tested in each of 12 basins distributed across CONUS over an eight year period from 2010-10-01 to 2018-09-30 (Figure 2). Because

running the analysis over all of CONUS is computationally prohibitive, these basins were selected to represent variability of NWM calibration basins over CONUS. Calibration basins minimize volume errors while the 12 basins span a wide range of climate, land cover, and terrain conditions.

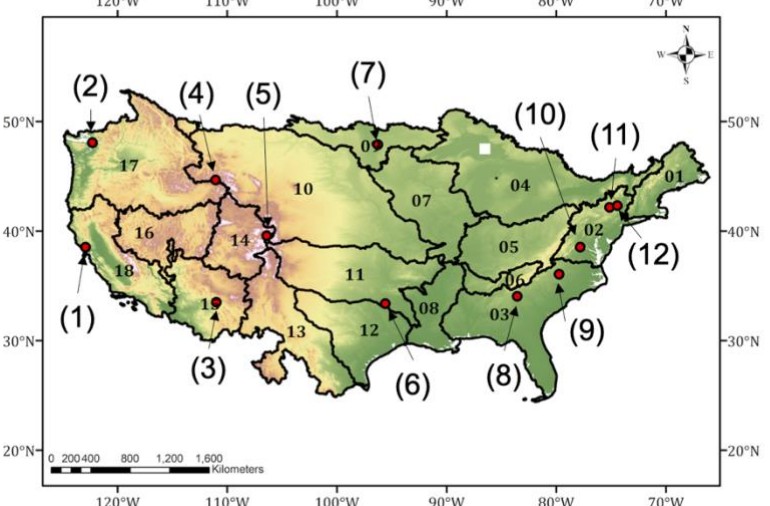

| Number | USGS Gage ID |
|--------|--------------|
| 1 | 11467000 |
| 2 | 12158040 |
| 3 | 09498502 |
| 4 | 06037500 |
| 5 | 09064600 |
| 6 | 07342500 |
| 7 | 05078500 |
| 8 | 02217475 |
| 9 | 02095000 |
| 10 | 01664000 |
| 11 | 01350000 |
| 12 | 01423000 |

**Figure 2: Map of study domains, with red points showing locations for 12 representative basins dispersed across CONUS used for the channel routing module sensitivity analysis. These numbers correspond to the USGS gage IDs listed in the table. Numbers and boundaries on the map correspond to the 18 designations and extents of HUC2 regions.**

 

A collection of output metrics describing model fit to observed data including normalized mean bias (NMB; Yu et
al., 2006), Nash-Sutcliffe Efficiency (NSE; Nash and Sutcliffe, 1970), and Richards-Baker flashiness index (R-B index; Baker
et al., 2004) were used to reduce the model output time series to scalar values more readily comparable to the input parameter
set. Equations for these metrics are provided below, as:

$$NMB = \frac{\sum_{i=1}^{N}(M_i - O_i)}{O_i} \times 100\% \tag{3}$$

$$NSE = 1 - \frac{\sum_{i=1}^{N}(M_i - O_i)^2}{\sum_{i=1}^{N}(M_i - \overline{O_i})^2} \tag{4}$$

$$R - B\ Index = \frac{\sum_{i=1}^{N}|M_i - M_{i-1}|}{\sum_{i=1}^{N} M_i} \tag{5}$$

where M is the model streamflow, O is observed streamflow, i is the time-step, and N is the total number of time-steps. The
optimal value for NMB is 0%, the optimal value for NSE is 1, and the optimal R-B Index is one which matches observations.
Normalized mean bias provides an unbiased, symmetric measure of tendency to overpredict or underpredict scaled by the
output flow values, NSE is a widely used measure of model goodness-of-fit to the overall observational time series, and R-B
flashiness evaluates how short-term changes in streamflow are affected by the channel routing parameterization. These metrics
were selected as they each provide unique insights into model performance and have previously been effectively used for
evaluation of hydrologic models in similar applications (e.g., Avellaneda and Jefferson, 2020; McInerney et al., 2018; Wu et
al., 2012; Yeste et al., 2020).

**2.4 Channel Parameter Development**

Channel parameters were first estimated at HyG-associated NHD reach segments, then subsequently estimated at all CONUS
river reaches, through a regression-based regionalization approach. The at-a-station hydraulic geometry of a channel (AHG)
was calculated by relating the cross-sectional variation of stream discharge with width, depth, and velocity using power law
relationships (Leopold and Maddock Jr., 1953), i.e.,

$$w = aQ^b \tag{6}$$
$$d = cQ^f \tag{7}$$
$$v = kQ^m \tag{8}$$

where w is width, d is depth, v is velocity, Q is discharge (equal to the product of w, d, and v), a, c, and k are fitted coefficients
which must multiply to 1, and b, f, and m are fitted exponents which must sum to 1. Using the field measurements of Q
available in HyG, we first estimated the fitted coefficients (a, c, and k) and exponents (b, f , and m) at each HyG location. For
a given flow percentile, variables w, d, and v were then calculated using the fitted values in Eqns. 6-8.





Manning's n was estimated using w, d, v, and longitudinal slope (S) via Equations 9 and 10, below:

$$R = \frac{w \times d}{w + 2d} \tag{9}$$

$$n = \frac{R^{2/3} \times S^{1/2}}{v} \tag{10}$$

where R is the hydraulic radius. Equation 10 is a version of the Manning's equation. Generally, longitudinal water surface slope is not measured at USGS and state stream gaging locations. Instead, values for slope were obtained from the NHDPlus dataset attribute "ElevSlope", a longitudinally smoothed slope product produced from topographic data (USGS, 2001).

For estimating the channel geometry parameters BW and m used in the NWM routing model, a 'half-channel'
conceptualization was used. For a given gage, field measurements of ½(w) and d together allow for calculation of the channel side slope. This was fit through a linear regression, e.g.,

$$d = m \times w + b \tag{11}$$

and the point where d = 0 along this fitted line is taken to be ½(BW). The slope of this fit is the channel side slope (m). TW was estimated as the width of the channel at a high percentile flow (e.g. 99th or 99.9th), which was analyzed through the model validation described in Sec. 2.6.

**2.5 Regionalization Analysis**

We conducted an analysis of the regionalization method using the Manning's n parameter as a representative for the full suite of channel parameters described in Figure 1 given the importance of roughness defined in prior studies. Manning's n was regionalized to unobserved channels in the stream network using a regression-based method. We fit linear regressions between log-transformed Manning's n and S at a flow percentile, *i*, as:

$$log\ (n_i)\ = m \times log\ (S_i)\ + b \tag{12}$$

where m is the slope of the regression line, and b is the intercept.

Training of the regionalization regression equations for Manning's n was performed at three spatial scales: HUC4, HUC2, and the full CONUS-wide domain. For each scale, only the observed data available within each spatial unit of that scale were used to estimate at reaches within that unit. The purpose of this multi-scale analysis was to attribute error in estimated Manning's n to variation in scale. In other words, maximization of available observations to fit the Eqn. 9 regression
and minimization of regression error are competing objectives, such that the scale which results in the least error may vary by location. Similarly, because Manning's n may vary based on the flow percentile used to estimate them (e.g., Eqn. 8), the regionalization was conducted using a range of flow percentiles at HyG locations, including the 50th, 75th, 90th, 95th, and 99th percentile flow values. The variation across three spatial scales and five flow percentiles results in a 3x5 matrix of estimated Manning's n values at HyG locations.



225         To facilitate a standardized method for evaluating the regionalization, a k-fold cross validation (CV) was performed using a value of k = 10 folds. In this approach, training data were randomly divided into ten equal-sized groups and systematically withheld one group at a time while training the model with the remaining nine. We then predicted Manning's n for the withheld group and compared regression-predicted values with the HyG-derived estimates.

**2.6 CONUS-wide Evaluation Experiments**

To understand regional- and national-scale implications of new channel parameters, the NWM routing module was run across the entirety of the 2.7 million CONUS reaches, over a period of 8 years from 2010-10-01 to 2018-09-30. As only the routing module was run (i.e., not the LSM), total channel inflow volumes remained fixed across experiments, such that any variation may be attributable to routing parameterization. Here, nine channel parameter set configurations were used in addition to the

v2.1 default configuration, for a total of ten experimental trials. These configurations included parameter sets with Manning's n regionalized at HUC4, HUC2, and full CONUS-wide domain spatial scales using 95th percentile flows. Channel geometry sets included default parameter values along with HUC4-scale regionalized estimates, with TW calculated using either the 99th ("TW99") or 99.9th ("TW99.9") percentile flows. This creates a 3x3 matrix of Manning's n and geometry combinations in addition to the default parameterization (Figure 3).






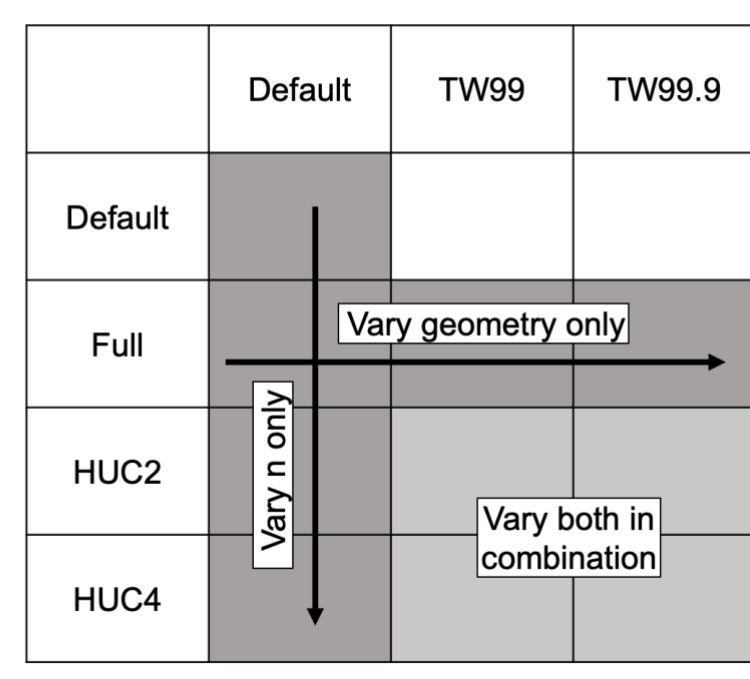

**Figure 3: Summary of Manning's n and channel geometry configurations used for the routing module simulations conducted across CONUS. Dark gray boxes indicate configurations where only Manning's n or channel geometry were updated from the default parameterization, and light gray boxes indicate configurations where both were modified.**


The experimental trials were evaluated with the objective of identifying whether errors affecting GOF metrics performance were reduced relative to the default configuration. This evaluative approach was used because the routing module only controls the flow routing through the system, rather than total flow volume. We compared the hourly streamflow output of each trial at observed reaches CONUS-wide using available gage observations, and also conducted a closer examination of

simulated flows for a selection of individual gages from the 12 representative basins (Figure 2). For the CONUS-wide analysis, GOF metrics such as percent bias (Eqn. 3), NSE (Eqn. 4), and $R^2$ were calculated at each stream gage. The difference between median experimental trial output metrics and default output metrics was used to quantify where and how updates to channel parameterization resulted in the greatest differences. The variance among experimental trial output metrics was also examined to further characterize agreement among trials.





## 3 Results

### 3.1 Sensitivity Analysis

Across all calculated metrics and domains, Manning's n was shown to hold the highest first order and total effect sensitivity indices, indicating a higher sensitivity of model output to Manning's n (Figure 4). The difference between Manning's n and other channel parameter sensitivities varied considerably across these dimensions, however. Normalized bias in comparison to gage observations showed increased sensitivity for other parameters relative to Manning's n across all basins, particularly BW and $n_{cc}$, for total effect sensitivity.

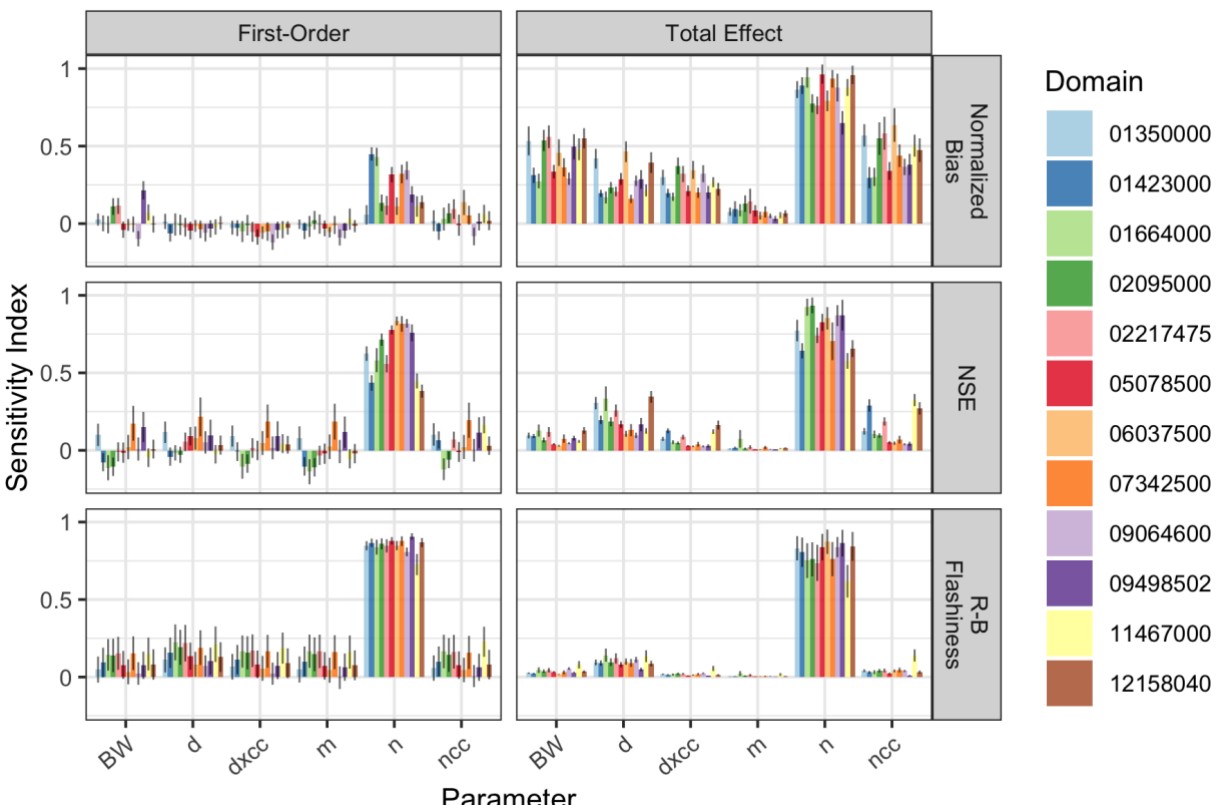

**Figure 4: Sensitivity analysis results for the NWM channel routing module in 12 representative basins (Figure 1). Estimated first order indices are listed in the left column, and total effect indices are in the right column. Indices closer to 1 indicate higher sensitivity, and values less than zero occur due to numerical instabilities within insensitive variables. Rows represent each of three metrics used to reduce flow time series to scalar values. Bar plot colors correspond to the domains for which the indices to apply.**





### 3.2 Channel Geometry Regionalization

The regression fit between log(S) and log($n_i$) varied by location and scale (Figure 5). The CONUS-wide log-transformed
regression fit with 5,777 observation points yielded an $R^2 = 0.29$. At the HUC2 regionalization level, $R^2$ varied from 0.12 in
the Texas-Gulf region (12) to 0.66 in the Great Basin region (16), with an overall median $R^2 = 0.37$ and median observation
count of 290 points in each HUC2. At the HUC4 level, variance was even greater, with $R^2$ falling between $4 \times 10^{-5}$ and 0.9 for
subregions 0302 and 0704, respectively. Overall, the median $R^2 = 0.45$ and median observation count was 26 points per spatial
unit. At this scale, there is also an apparent east to west gradient of decreasing error, largely due to the presence of low $R^2$
HUC4 basins in the Texas-Gulf (12) and South Atlantic-Gulf (03) regions. Kernel density plots for error in Manning's n
subdivided by regionalization scale and HUC2 region are shown in supplementary materials (Figure S1).

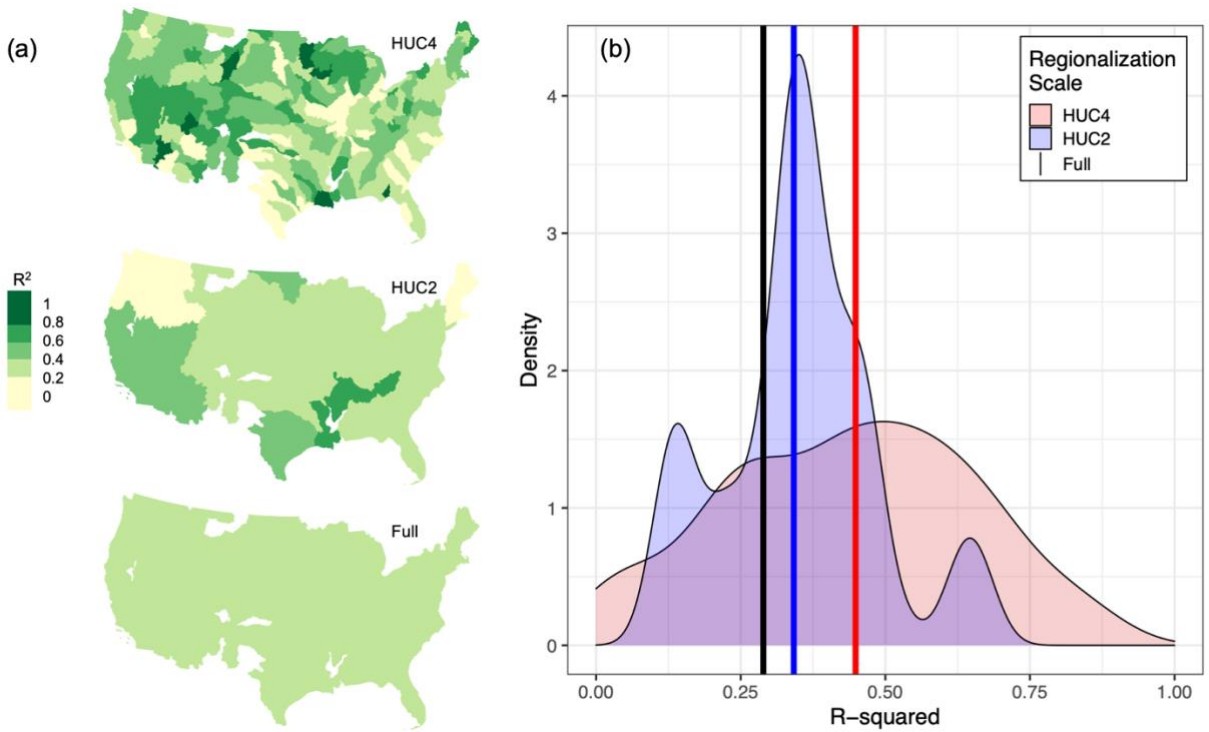

**Figure 5: Summaries of R2 values resulting from the regression fit of Manning's n as a function of channel longitudinal slope at
HUC4, HUC2, and full CONUS domain regionalization scales. Panel (a) shows a spatial breakdown of R2 values at units within each
scale, and Panel (b) shows kernel density plots for regressions made at the HUC4 and HUC2 scales, with vertical lines denoting the
R2 from the full CONUS-wide fit, and the median values for the HUC4 and HUC2 scales.**

Cross validation results from the 3x5 matrix of regionalization scale and flow percentile combinations are summarized
in Figure 6. General patterns of decreasing error with increasing flow percentile and finer scale were evident, with some
exceptions. For example, the regression determined from 90th percentile flow yielded the smallest Manning's n error in the



California region (18), whereas the smallest error in the Tennessee region (06) was achieved at the full CONUS-wide regionalization scale. However, channels in mountain west HUC2 regions (e.g., 13-17) were poorly represented by the

regression made at the full CONUS-wide scale, as evident by the relatively strong underestimation of Manning's n.

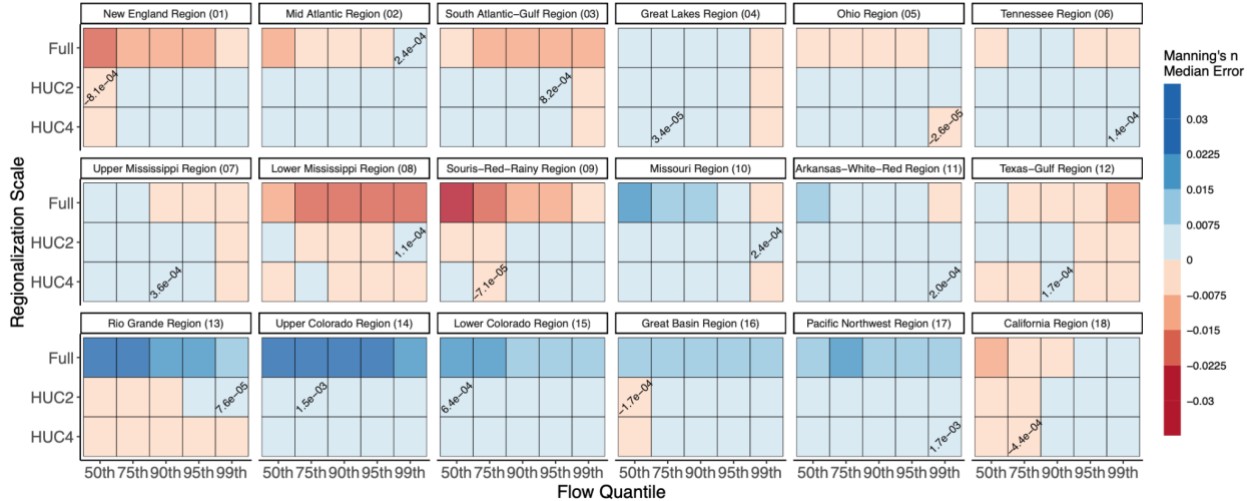

**Figure 6: A summary of median error in Manning's n resulting from a k-fold cross validation (k=10) across the matrix of tested regionalization scale and flow percentile combinations. HUC2 regions are shown in each facet, and boxes with text indicate the**
**combination that resulted in the lowest error, which is shown within the box.**

Overall, nearly half of the HUC2 regions (8 of 18) showed the 99th percentile as the optimal flow percentile. However, the optimal regression fit was relatively balanced between the HUC2 and HUC4 regionalization scales, with eight regions minimizing error at HUC2 scale, and nine regions minimizing error at the HUC4 scale. Variability in error was highest in the

Lower Mississippi region (08) where the ratio between slope and Manning's n varied greatly among observed locations, and there were fewer observations (Figure S2). In western regions (e.g., 13-17), a strong positive bias in estimated Manning's n at the full CONUS-wide regionalization scale was evident.

### 3.3 CONUS-Wide Channel Parameters

In comparison to the default parameterization, the experimental parameter combinations described in Figure 3 resulted in substantial differences to both estimated Manning's n and channel cross-sectional area (Figure 7), with the HUC4 regionalization and TW99.9 geometry configuration presenting the greatest differences from default. The majority of channels (76%) are represented in the default NWM version by a Manning's n value of 0.06. The regionalized Manning's n updated these values to a new range between 0.006 and 0.537 (median = 0.077), most noticeably in mountainous headwaters regions,

where roughness increased by approximately 200% under the HUC4 regionalization scheme. Similar changes were also





apparent under the HUC2 (0.007 to 0.436, median = 0.076) and full regionalizations (0.012 to 0.436, median = 0.072), albeit to a lesser extent. Overall, the variance of Manning's n across CONUS increased with smaller regionalization spatial scales. Similar magnitudes of change were evident in channel cross-sectional area. Compared to the default geometry parameterization cross-sectional area (0.018 m$^2$ to 1990 m$^2$, median = 2.03 m$^2$), the TW99 configuration (8.53 x 10$^{-7}$ m$^2$ to 8610 m$^2$, median =

0.927 m$^2$) and TW99.9 configuration (1.62 x 10$^{-4}$ m$^2$ to 7150 m$^2$, median = 2.08 m$^2$) both resulted in wider ranges, though median area for TW99 was reduced by an order of magnitude, and this reduction was largely observable at reaches located across the West. However, in the Lower Mississippi region (08), the cross-sectional area of the channel increased by approximately 200% under the new regionalization schemes.

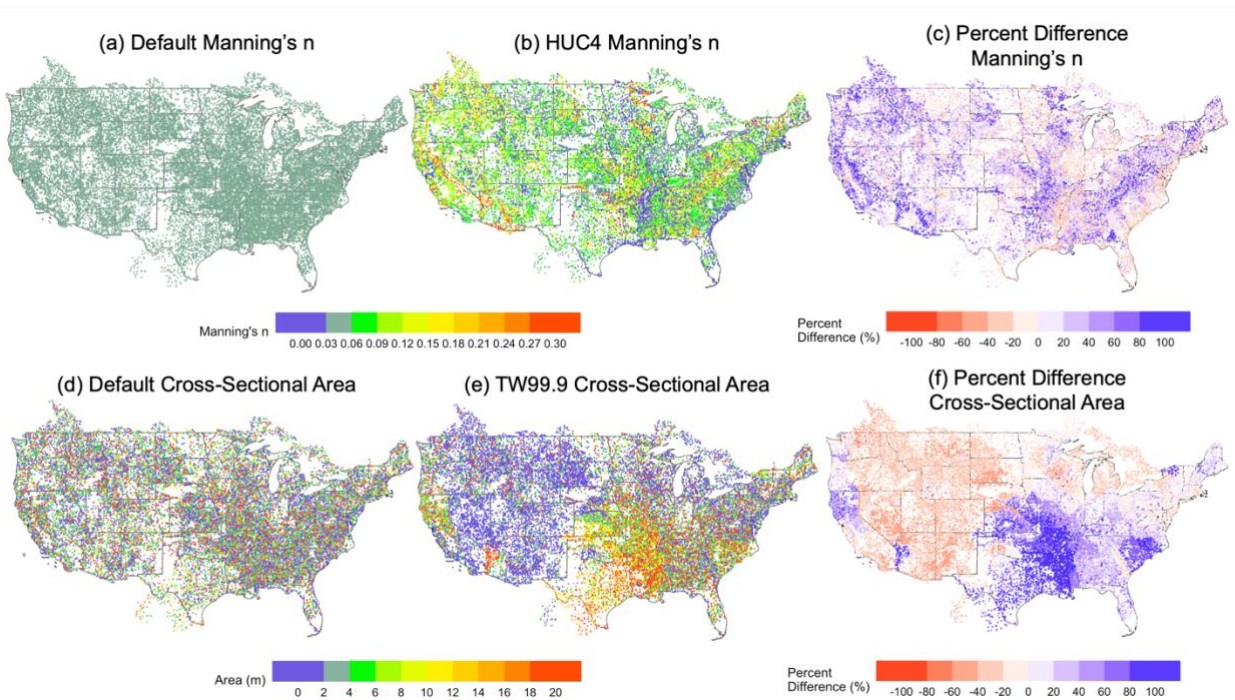


**Figure 7: Spatial maps illustrating default parameterizations, updated parameterizations, and their percent differences across CONUS for Manning's n and channel geometry-derived cross-sectional area at a random subsample of 1% of the 2.7 million NHD-derived reaches across CONUS. Panels (a) and (b) describe the default and HUC4-regionalized Manning's n parameter, panels (d) and (e) describe the default and TW99.9 configuration cross-sectional area, and panels (c) and (f) show the percent differences**

**between default and updated parameterizations for Manning's n and cross-sectional area, respectively.**

Across the 6,841 USGS gage locations with continuous information across the experimental period, the median percent difference between default Manning's n and the HUC4-regionalized Manning's n was approximately -9% with a standard deviation $\sigma$ = 61% (HUC2: median = -8%, $\sigma$ = 52%; full: median = -10%, $\sigma$ = 49%). For the channel cross-sectional

area using 99th percentile flow to estimate top width (TW99), this difference was -32% with a standard deviation $\sigma$ = 47%





(TW99.9: median = 17%, $\sigma$ = 88%). Generally, median channel size was reduced in the TW99 configuration and increased in the TW99.9 configuration (Figure 8).

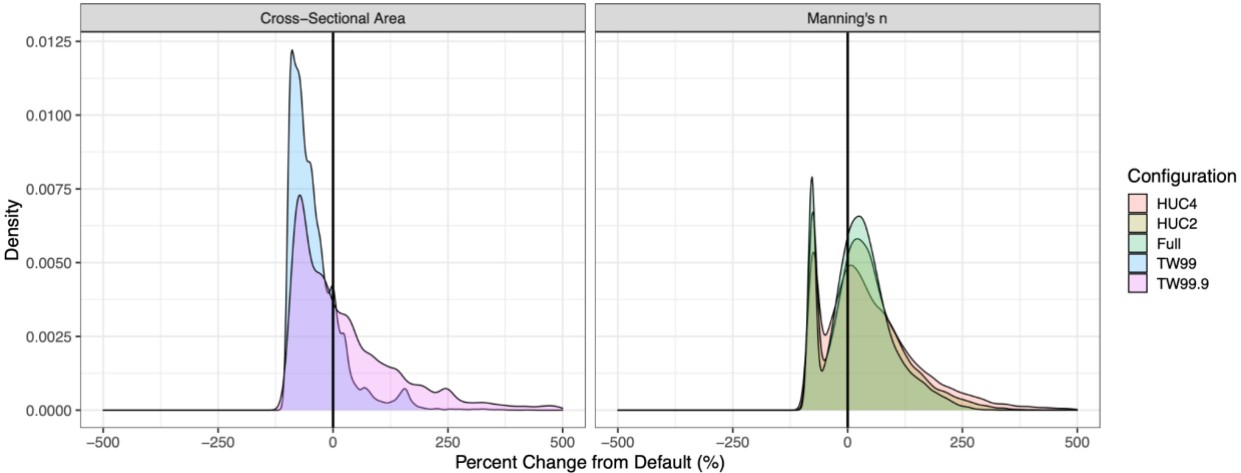

**Figure 8: Density plots of percent changes in cross-sectional area and Manning's n from default parameter values across CONUS under experimental trial configurations.**

## 3.4 CONUS-Wide Evaluation Experiments

Among the experimental trials, variance was generally low (approx. 1 x 10⁻⁵) in the bulk goodness-of-fit metrics calculated
from the model output at gage locations, indicating little difference in model output among the updated channel parameter sets resulting from modifying routing module parameterization alone. Yet, differences between median experimental trial output metrics and the default output metrics yielded some measurable differences, particularly for the agreement index ($\sigma$ = 2.8 x 10⁻²) and R-squared ($\sigma$ = 4.5 x 10⁻²) metrics. Effects on performance were negligible across most gages, with the median value for each metric approximately zero in all cases (Figure 9). Spatial maps for other metrics are provided in Figure S3.


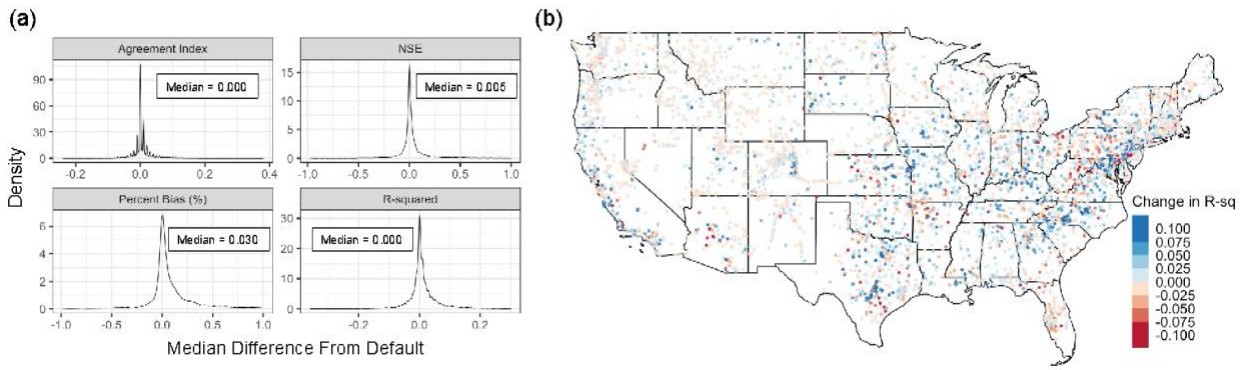





**Figure 9: Summary metrics at 4,655 USGS gage locations for the HUC4 regression scale, TW99.9 combination experimental trial calculated across gage locations CONUS-wide. Panel (a) shows faceted density plots for each difference in that metric from the default parameterization, and panel (b) shows the Pearson correlation among metrics.**


Overall, mean $R^2$ across gages increased from the default parameterization (mean $R^2 = 0.479$) for all experimental trials (from a mean $R^2 = 0.489$ for the Full TW99 configuration to a mean $R^2 = 0.494$ for the HUC4 TW99.9 configuration). The HUC4 TW99.9 configuration also resulted in the largest overall influence on model output overall (i.e., regardless of whether performance improved or worsened), which is consistent with the degree of perturbation made to the channel
parameters relative to other configurations.

### 3.5 Analysis at Selected Gages

Of the 12 representative basins (Figure 2), two gages at outlets were selected for further examination based on the relatively high degree and opposing directions of change made to the parameterization of Manning's n and channel geometry relative to the default parameterization, as well as differences in basin physiography and climate. The first is USGS gage 09064600 at
Eagle River near Minturn, CO. This gage is located in the mountainous headwaters region of the Colorado River basin (elevation 2,467m), and monitors flow over a drainage area of 482 km². From a default value of 0.055, Manning's n was increased for this gaged reach to 0.078, 0.073, and 0.097 for HUC4, HUC2, and full regionalization scales, respectively. The cross-sectional area was reduced from a default of 19.9 m² to 4.9 m² and 6.4 m² for TW99 and TW99.9 configurations, respectively. The second is USGS gage 01664000 at Rappahannock River near Remington, VA. This gage is located at a lower
elevation (92m) and monitors flow over a greater drainage area of 1,603 km². In contrast to the Colorado gage, the default value of 0.050 for Manning's n was decreased for this channel to 0.017, 0.015, and 0.015 for HUC4, HUC2, and full regionalization scales, respectively. The cross-sectional area was altered from a default of 37.5 m² to 35.4 m² and 65.7 m² for TW99 and TW99.9 configurations, respectively.

Noticeable differences in the behavior across experimental scenario results exist between the two selected gages
(Figure 10). While NSE, $R^2$, and RMSE were relatively consistent across experimental trials for gage 09064600, there is a noticeable trend of decreasing performance from the default parameterization run and updated Manning's n runs when channel geometry is updated. The highest differences across experiments were those where channel geometry alone was perturbed: NSE was 0.50 for the default parameterization, and 0.51, 0.44, and 0.41 for the default geometry with updated Manning's n only, the TW99 channel geometry parameterization, and the TW99.9 channel geometry parameterization, respectively. By
contrast, experimental performance among trial where only Manning's n was perturbed were relatively consistent and higher than default parameterization for both gages. For gage 09064600, $R^2$ increased from the default parameterization ($R^2 = 0.77$) for all runs, with the highest increase for the HUC4 TW99.9 run ($R^2 = 0.81$).





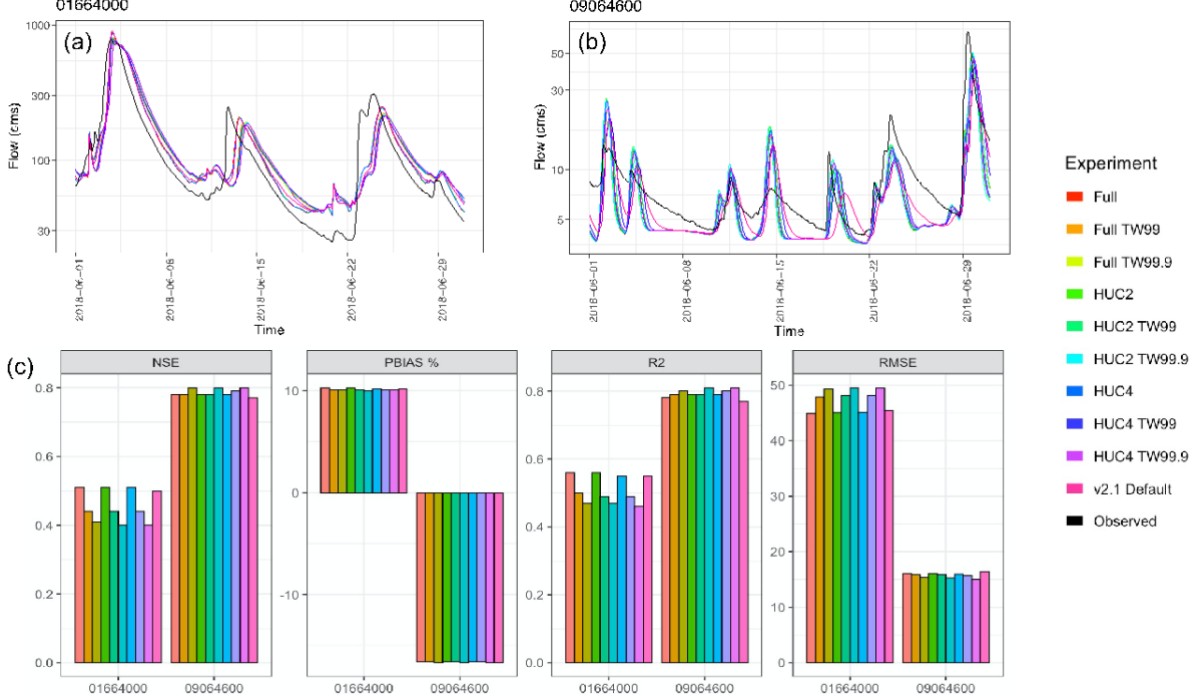

**Figure 10: A summary of experimental results at two gage locations. Panels (a) and (b) describe hydrographs for USGS Gage 09064600 and USGS Gage 01664000, respectively, for one example month in June 2018, and panel (c) shows metric results computed across the entire hydrograph at each gage location for NSE, percent bias, R2, and RMSE.**

## 4 Discussion

Results from the sensitivity analysis showed that channel roughness (Manning's n) holds a stronger influence on modeled streamflow than channel dimensions in the routing module. This finding is supported by prior literature suggesting that Manning's n is a significant determinant of flood wave celerity (Anderson et al., 2006) and serves to attenuate and delay the arrival of peak discharge at the catchment outlet (Wolff and Burges, 1994; Woltemade and Potter, 1994). In the NWM, attenuation is modeled through the Muskingum-Cunge method of flood routing (Cunge, 1969), which uses a diffusion wave representation subject to attenuation as it propagates through a channel network. The relative sensitivity of model output variance to Manning's n suggests that the most efficient optimization strategy for improving representation in the channel routing module is one that is focused on updating Manning's n. However, the overall results from the experimental simulations showed that runs where channel geometry was varied in isolation generally yielded a higher variance in model output than runs where Manning's n was varied in isolation. Such results show that combinatory effects among the channel geometry parameters may result in a stronger influence over the model hydrograph in comparison with varying the Manning's n parameter alone. The case for updating all geometry parameters is strengthened by the fact that the HUC4 TW99.9 configuration, containing the most extreme perturbations of all parameters, was found to increase $R^2$ by the largest degree.





Though the results were largely conclusive, several aspects of the sensitivity analysis may be modified in potential future studies. For example, the selected boundary conditions ranged from 0.1 to 10 times the nominal parameter values, which may not be reflective of the true uncertainty in these parameter values. The possibility also exists that parameter sensitivity is

flow dependent, which may be most obvious in the case of the parameters $TW_{cc}$ and $n_{cc}$, as flow depth is often too low to reach the floodplain. Consideration for observational error in channel parameters and/or running the model in data assimilation mode may address this possibility and provide added value in future analyses.

The regionalization of channel parameters was performed using a HUC-based approach, where discrete regions were used to define the regression curves used to estimate channel parameters within those regions. The principal finding in

comparing regionalization scales was that a smaller scale typically results in the lowest error (e.g. Figure 6), and the magnitude of this difference is likely dependent on the inherent spatial variability of the region in which the regressions were developed. For example, the relatively poor performance of the full regression in the topographically variable, mountainous HUC2 regions demonstrate the non-representativeness of regressions developed using all measurements across CONUS for these unique and topographically complex areas. Furthermore, the strong performance of the HUC4 regionalization scale relative to HUC2 in

the Missouri Region (10) speaks to the diversity of terrain conditions within this region, as it encapsulates both mountainous terrain in the west and flatter plains in the east. Overall, these findings underscore the importance of taking into account the spatial variability of the Manning's n and longitudinal slope relationship. Additional variables which demonstrate strong relationships with channel properties may also be viable for future regressions. For example, height above nearest drainage (HAND) has been used to derive hydraulic properties for reaches along a river network and generate synthetic rating curves

relating flow to water level (Zheng et al., 2018).

The HUC-based discretization method, coupled with differences in observational data uncertainty and availability, naturally creates discontinuities at HUC boundaries in the regression parameters. Alternative regionalization approaches may help to alleviate or even remove the errors arising from these discontinuities. For example, a downstream hydraulic geometry (DHG) based regionalization approach that takes into account observational data from nested gages within the network to

generate channel parameters along a flow path is one possibility that has seen previous success (Allen et al., 2018; Neal et al., 2015).

The estimation of regional regression curves for Manning's n was performed across multiple flow percentiles, as it was found that the regression parameters varied depending on flow. Here, the objective was to identify a singular optimal flow percentile that resulted in the lowest error in the regionalized Manning's n parameter. However, in nature, the celerity and

attenuation of a flood wave varies nonlinearly with flow, despite standard engineering practice typically involving the use of the Muskingum-Cunge flood wave representation due to ease of implementation (i.e., a constant Manning's n), which is the case for the NWM. Future improvements to the NWM may consider allowing Manning's n to vary with flow, as this may achieve better representation of channel hydraulics.

With only a modest sensitivity of model output to channels parameters, results from both the sensitivity analysis and

simulations demonstrate the limited influence of the channel routing module to improve goodness-of-fits metrics within the





overall NWM framework. In most cases, low variability in GOF metrics among trials is evident, though in some instances, such as at USGS gage 09064600, there is some identifiable improvement from the default parameterization. Yet, even here, model hydrographs were unable to match observations. This is expected, as total volume is unaffected by the routing module, and thus mass is conserved regardless of channel parameterization. However, in the course of model improvement, an
appropriate philosophy is to 'do no harm', which largely characterizes the outcome of these experiments.

Parameters within the Noah-MP LSM not included in the sensitivity analysis or regionalization are likely the source of a large percentage of error, with meteorology and physics representation representing other potential sources. A previous sensitivity analysis conducted on the Noah-MP model indicated high sensitivities for output states and fluxes such as sensible and latent heat, soil moisture, and net ecosystem exchange derived from soil and vegetation parameters (Arsenault et al., 2018).
Another showed sensitivity for latent heat and total runoff attributable to two-thirds of applicable standard parameters, and the highest sensitivity derived from a hard-coded parameter value in the model used in the formulation of soil surface resistance for direct evaporation (Cuntz *et al.*, 2016). Given these results, future efforts focused on joint calibration of the Noah-MP LSM and channel routing module may result in noticeable GOF metrics improvements.

## 5 Conclusion

This analysis explored the effects of modifying channel routing parameters in the National Water Model streamflow simulations using a regionalized hydraulic geometry and Manning's n dataset. Based on a sensitivity analysis conducted on a selection of channel parameters in the routing module, it can be concluded that the Manning's n roughness coefficient holds an outsized effect on modeled flow relative to parameters which describe the channel geometry. Yet, results from experimental simulations of nine alternative parameter configurations showed the interactive effects among geometry parameters in some
geographic regions may be greater than the Manning's n parameter alone.

New estimates of NWM channel parameters following a regression-based regionalization approach generally results in a larger distribution of channel characteristics over the NWM v2.1 default parameterization. Overall, variance in both Manning's n and cross-sectional area among channels CONUS-wide increased from the default parameterization, which also accompanied a modest increase to median $R^2$ across gage locations as well, from 0.479 to 0.494 for the HUC4 TW99.9
configuration.

For Manning's n, approximately 76% of channels in the default parameterization are currently represented by the same nominal value of 0.06 (18% with a value of 0.055, and lesser percentages at further intervals of 0.005), and based not on observations but rather expert opinion, scaled by Strahler stream order. A new HyG-based Manning's n representation provides an observational foundation for Manning's n, which consequently increases roughness across mountainous headwaters regions
and decreases roughness in lowlands and coastal areas to a new range between 0.006 and 0.537 (median 0.077), qualitatively changing the distribution.



Channel geometry updates resulted in a longitudinal gradient in percent change in cross-sectional area. In the East, and particularly in the Lower Mississippi region, cross-sectional area increased, while a decrease in area is visible throughout smaller streams in the more arid West.

The influence of the routing module over modeled streamflow GOF metric performance is limited compared to other components of the NWM framework, such as the land surface model and meteorological input data. Future approaches towards calibration of the NWM may yield the largest benefits through a more holistic approach to calibrating the overall framework, i.e., comprehensive evaluation and calibration of all model components. Towards this objective, our characterization of the overall effects of strengthening channel routing module parameter representativeness may serve as an important foundation

for further improvement of the NWM and hydrologic modeling in CONUS. In turn, the NWM becomes better positioned to meet the stated goal of providing quality, actionable guidance for mitigation of flood-related damages.

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


## Appendix A

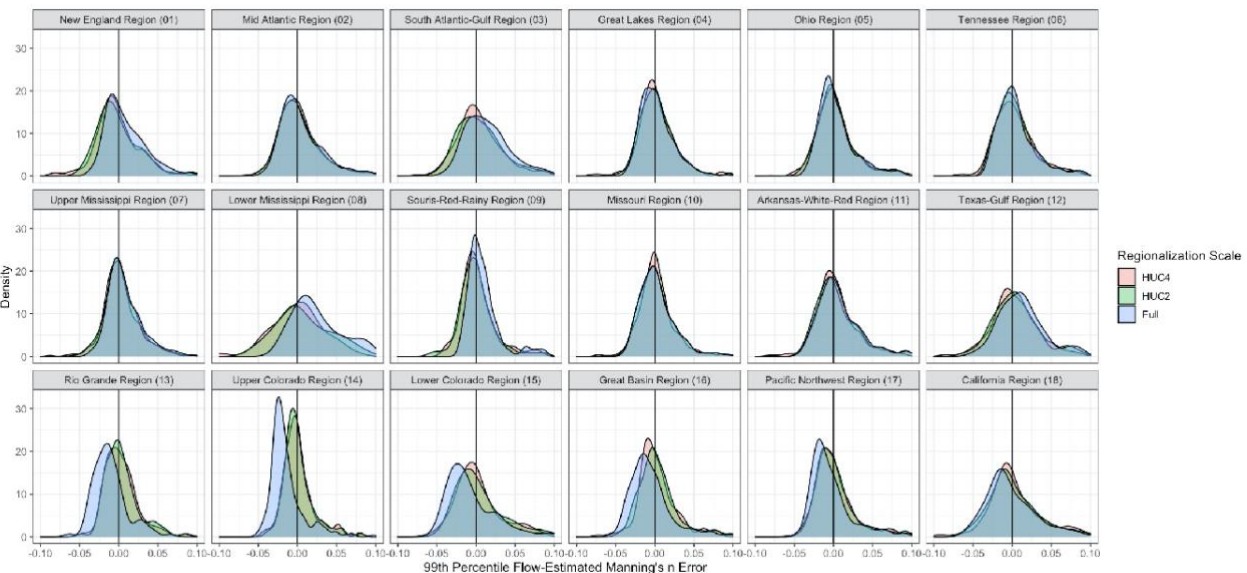

**Figure A1: Kernel density plots showing the range of Manning's n error resulting from each regression-based regionalization scale**
600 **at gage locations, including full CONUS-wide (blue), HUC2 (green), and HUC4 (red). Facets indicate the HUC2 region in which the gages are located.**

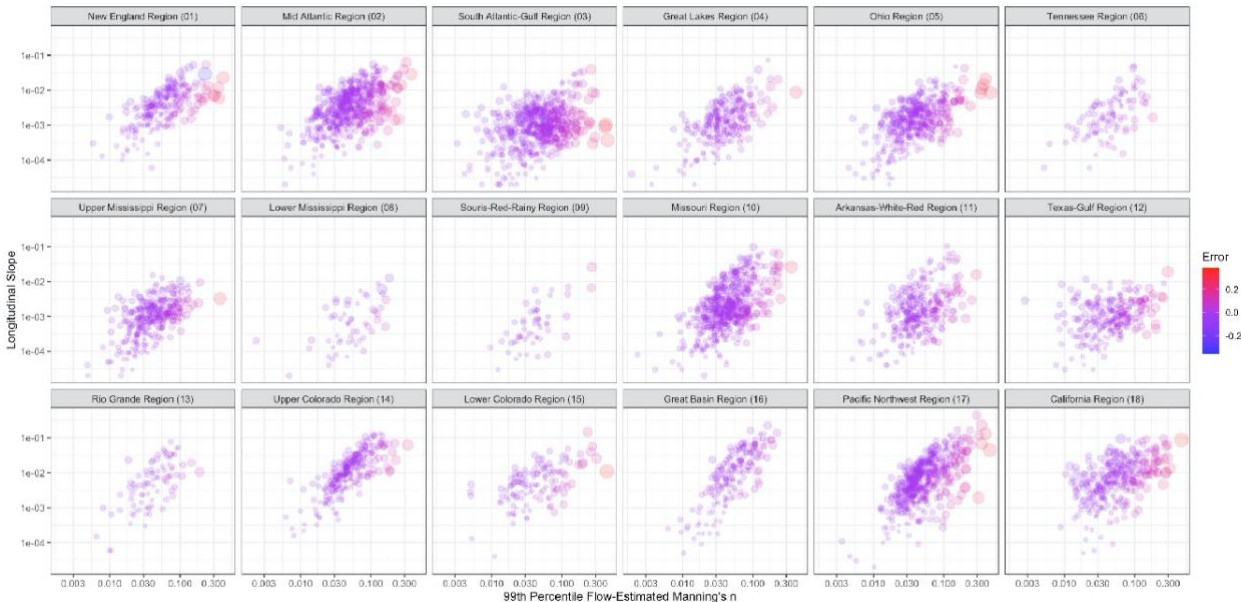

**Figure A2: Scatterplots of log-transformed longitudinal slope (S) and Manning's n (n) estimated at 99th percentile flows for HyG**
605 **locations in each HUC2 region are shown. Size of the points indicate the magnitude of error in the regression, and color indicates an underestimate (blue) or overestimate (red).**





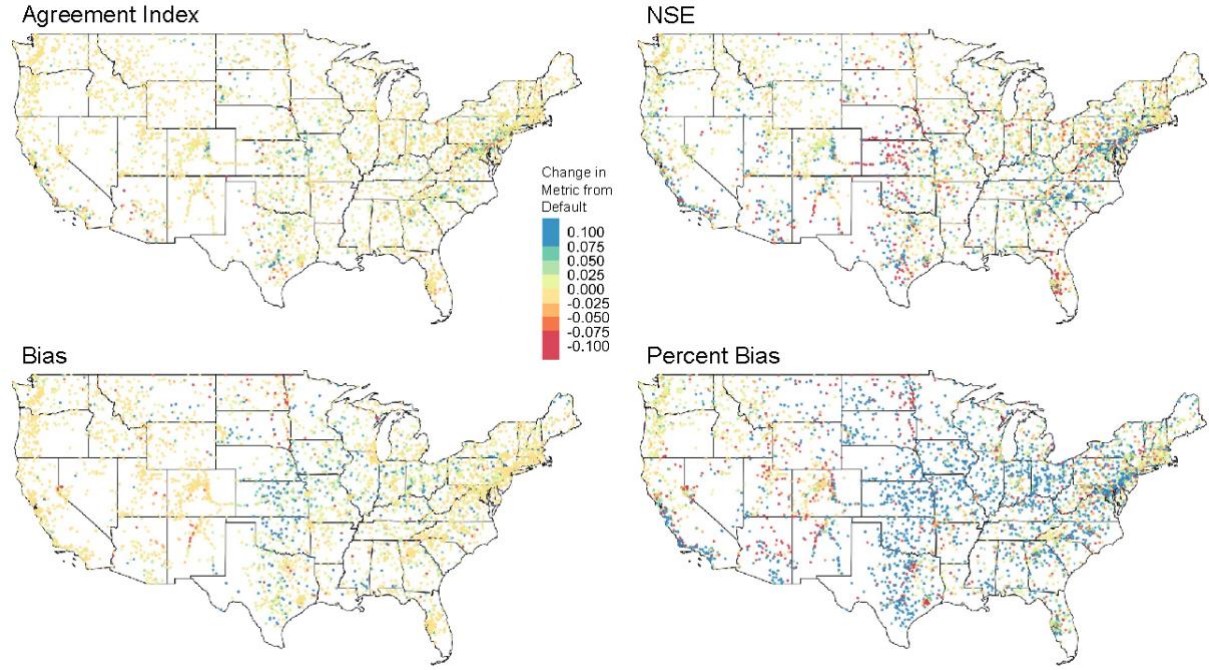

**Figure A3: Spatial maps describing change in Agreement Index, NSE, bias, and percent bias from default parameterization performance at USGS gage locations across CONUS.**