# Peer review of "Evaluation of a New Observationally Based Channel Parameterization for the National Water Model"

_Hydrology and Earth System Sciences, 2021_

## Referee Comment (RC2)

[referee-annotated manuscript omitted]

---

## Author Comment (AC1)

**Response to Anonymous Referee #1**

In this paper, the authors explored the effects of modifying channel routing parameters in the National Water Model (NWM) streamflow simulations using a regionalized channel geometry and Manning's roughness dataset. The study is an important contribution to the possibility of improving the NWM in order to provide a better quality of the results, focusing especially on those areas where significant differences were found. This reviewer considers the paper is suitable for publication in Hydrology and Earth System Sciences" Journal, after the authors address the following suggestions and comments.

*We thank the reviewer for their acknowledgement of this manuscript's value and provide responses to each comment in order of appearance.*

Line 22. An increase in mean R2 by just over one hundredth (from 0.479 to 0.494) is not a significant variation. I suggest changing the word "significant" for "modest" as you do in the Conclusions (Line 454).

*While we characterize the improvement in mean $R^2$ as "statistically significant" in the abstract, we acknowledge the reviewer's point that the improvement is indeed modest and make the suggested change for clarity.*

*Original, Line 22: "…with a statistically significant mean $R^2$ increase from 0.479 to 0.494 across approximately 7,400 gage locations."*

*Revised, Line 22: "…with a **modest** mean $R^2$ increase from 0.479 to 0.494 across approximately 7,400 gage locations."*

Line 40. Here you must define the abbreviation "LSM" (Land Surface Model), since is the first time in the document that you are using it.

*We make this correction as suggested. We also remove the definition of LSM which occurs a few lines later.*

*Original, Line 40: "This research is focused on the NWM channel routing module, and therefore does not investigate parameterization of the LSM, the gridded routing module, or any other component of the NWM framework."*

*Revised: "This research is focused on the NWM channel routing module, and therefore does not investigate parameterization of the **Land Surface Model (**LSM), the gridded routing module, or any other component of the NWM framework."*

*Original, Line 42: "In many cases, these data are produced by land surface models (LSMs) continuously forced by weather forecast data."*

*Revised: "In many cases, these data are produced by **LSMs** continuously forced by weather forecast data."*

Line 130. I suggest including a brief discussion on the dependence and variation of the parameter ncc as a function of the extension of the floodplain (dxcc variation). Same for the dependence and variation of n as a function of the channel depth (d variation).

*We will add some additional context noting the variation of n as a function of depth in both the main channel and the floodplain.*

Line 171. Explain more details about the criteria that you considered to define the 12 basins of study. It could be summarized in a table that list the climate, land cover and terrain characteristics for each basin.

*The selected basins are designated calibration basins for the NWM and have historically been used as testbeds for model improvement. We provide more detail on the basin selection process below.*

*Original, Line 171: "Because running the analysis over all of CONUS is computationally prohibitive, these basins were selected to represent variability of NWM calibration basins over CONUS. Calibration basins minimize volume errors while the 12 basins span a wide range of climate, land cover, and terrain conditions."*

*Revised: "Because running the analysis over all of CONUS is computationally prohibitive, these basins were selected to represent variability of NWM calibration basins over CONUS. **The** calibration basins **have historically been used as testbeds for model improvements and were selected based on such criteria as: 1) basin size maximum of 10,000 km², 2) availability of streamflow observational data, 3) a minimal basin disturbance index, and 4) presence of lakes in the basin (**RafieeiNasab** et al., 2020). The selected basins** minimize volume errors while span**ning** a wide range of climate, land cover, and terrain conditions."*

Line 201. One of my main concerns is that the authors didn't explore the uncertainty of the longitudinal slope (S) in Equation 10 to obtain Manning's n. The authors made two strong assumptions that should to be better discussed and justified: 1) Authors used the terrain slope instead of the hydraulic grade line in the Manning equation, and 2) the slope was not measured but obtained from a terrain model that could include errors of the DEM and those errors propagate in the obtained Manning's n.

*We appreciate the reviewer's concern regarding uncertainties that arise from the use of longitudinal slope in Manning's equation. Lacking direct observations of channel bed slope, we derive terrain slope as an approximation for this variable. We also assume uniform flow conditions for our experiment, which in turn assumes that the slope of the hydraulic grade line (i.e., the water surface slope in open channel flow) is equal to the channel bed slope. Without direct observations of channel bed slope, it becomes difficult to quantify the uncertainty that arises from these assumptions. We clarify our assumptions and note this uncertainty exists in our revision.*

*Original, Line 201: "Generally, longitudinal water surface slope is not measured at USGS and state stream gaging locations. Instead, values for slope were obtained from the NHDPlus dataset attribute "ElevSlope", a longitudinally smoothed slope product produced from topographic data (USGS, 2001)."*

*Revised: "Generally, longitudinal water surface slope is not measured at USGS and state stream gaging locations. Instead, values for slope were obtained from the NHDPlus dataset attribute "ElevSlope", a longitudinally smoothed slope product produced from topographic data (USGS, 2001).* **Implicit in this methodology are two assumptions: longitudinal slope is an adequate approximation of channel bed slope, and flow conditions are uniform causing water surface slope and channel bed slope to be equal.***"*

Line 206. I suggest renaming the parameter "b" in the linear regression Equation 11, as it might be confusing with the exponent "b" in Equation 6.

*We agree with the viewer's suggestion and replace the intercept variable "b" in Eq. 11 and 12 with the variable "$\beta_0$". For consistency, we also replace the variable "m" with "$\beta_1$":*

*Original, Line 207: "$d = m \times w + b$"*

*Revised: "$d = \boldsymbol{\beta_1} \times w + \boldsymbol{\beta_0}$"*

*Original, Line 214: "$\log(n_i) = m \times \log(S_i) + b$"*

*Revised: "$\log(n_i) = \boldsymbol{\beta_1} \times \log(S_i) + \boldsymbol{\beta_0}$"*

Line 235. Explain more details of the reasons you considered for choosing the 99th and 99.9th percentile flows to calculate TW.

*We provide additional context for this decision as follows:*

*Original, Line 235: " Channel geometry sets included default parameter values along with HUC4-scale regionalized estimates, with TW calculated using either the 99th ("TW99") or 99.9th ("TW99.9") percentile flows."*

*Revised: " Channel geometry sets included default parameter values along with HUC4-scale regionalized estimates, with TW calculated using either the 99th ("TW99") or 99.9th ("TW99.9") percentile flows.* **In the absence of observed top width data, the $99^{th}$ and $99.9^{th}$ percentile flows provide an estimate of TW such that the effects of perturbing the main channel geometry may be more consistently compared across flow volumes without the complex additional effects of overbank flooding that introduce uncertainty.***"*

Line 277. In this line you mentioned Figure S1 but in the Appendix A the Figure is called Figure A1. This figure also has a poor resolution and is difficult to read it. Improve the quality of the figure.

*We improve the quality of this figure as the reviewer suggests. We also correct the naming error in the manuscript by replacing S1 with A1.*

*Original, Line 277: "Kernel density plots for error in Manning's n subdivided by regionalization scale and HUC2 region are shown in supplementary materials (Figure S1)."*

*Revised: "Kernel density plots for error in Manning's n subdivided by regionalization scale and HUC2 region are shown in supplementary materials (Figure **A1**)."*

Line 287 - 289. The analysis written in these lines does not correspond to what is shown in Figure 6. The caption of Figure 6 says "…and boxes with text indicate the combination that resulted in the lowest error, which is shown within the box". However, you mention "For example, the regression determined from 90th percentile flow yielded the smallest Manning's n error in the California region (18), whereas the smallest error in the Tennessee region (06) was achieved at the full CONUS-wide regionalization scale" but according to Figure 6 for region 18 the smallest Manning's n error is in the 75th percentile and for region 6 the smallest error is for the HU4 scale. Review both the figure and the discussion for consistency.

*We thank the reviewer for noticing this discrepancy and correct it in the text.*

*Original, Line 287: "For example, the regression determined from 90th percentile flow yielded the smallest Manning's n error in the California region (18), whereas the smallest error in the Tennessee region (06) was achieved at the full CONUS-wide regionalization scale."*

*Revised: "For example, the regression determined from **75th** percentile flow yielded the smallest Manning's n error in the California region (18), whereas the smallest error in the **Mid Atlantic** region (**02**) was achieved at the full CONUS-wide regionalization scale."*

Line 302. In this line you mentioned Figure S2 but in the Appendix A the Figure is called Figure A2. Improve the quality of the figure.

*We correct this error in the manuscript by replacing S2 with A2.*

*Original, Line 302: "Variability in error was highest in the 300 Lower Mississippi region (08) where the ratio between slope and Manning's n varied greatly among observed locations, and there were fewer observations (Figure S2)."*

*Revised: "Variability in error was highest in the 300 Lower Mississippi region (08) where the ratio between slope and Manning's n varied greatly among observed locations, and there were fewer observations (Figure **A2**)."*

Line 344. In this line you mentioned Figure S3 but in the Appendix A the Figure is called Figure A3.

*We correct this error in the manuscript by replacing S3 with A3.*

*Original, Line 344: "Spatial maps for other metrics are provided in Figure S3."*

*Revised: "Spatial maps for other metrics are provided in Figure **A3**."*

Line 345. Improve the quality of Figure 9

*We improve the quality of this figure as the reviewer suggests.*

Line 380. Improve the quality of Figure 10

*We improve the quality of this figure as the reviewer suggests.*

Line 405. How can you ensure that a smaller scale typically results in the lowest error, if according to Figure 6, only 9 out 18 regions show the smallest Manning's error for HUC4 which is equivalent to 50% of the study regions, and 8 regions show the smallest error for HUC2 which is around 45%? I do not see much of difference here to affirm that sentence.

*The assertion that a smaller scale typically results in lower error is made from the perspective that both HUC2 and HUC4 regionalization scales typically result in lower error than the CONUS-wide regionalization scale, and that the HUC4 scale results in slightly less error than the HUC2 scale. There is a clear pattern in decreasing error with decreasing scale and it is something we believe is worth noting in the discussion, though we do acknowledge the modest differences between the HUC2 and HUC4 scales.*

*Original, Line 405: "The principal finding in comparing regionalization scales was that a smaller scale typically results in the lowest error (e.g. Figure 6), and the magnitude of this difference is likely dependent on the inherent spatial variability of the region in which the regressions were developed."*

*Revised: "The principal finding in comparing regionalization scales was that a smaller scale typically results in the lowest error (e.g. **the larger errors at the CONUS-wide regionalization scale relative to the HUC2 and HUC4 scales shown in** Figure 6), and the magnitude of this difference is likely dependent on the inherent spatial variability of the region in which the regressions were developed."*

Line 410. In your analysis you mention "Furthermore, the strong performance of the HUC4 regionalization scale relative to HUC2 in the Missouri Region (10) speaks to the diversity of terrain conditions…" However, according to Figure 6, in region 10 the smaller Manning's error was found in HUC2 which means the strong performance here is not for HUC4 regionalization scale.

*We acknowledge this discrepancy and remove this erroneous sentence from the manuscript.*

---

## Author Comment (AC2)

**Response to Anonymous Referee #2**

Overall the paper follows sound and well known techniques to estimate, regionalize, and calibrate channel geometric and friction parameters in applications used in continental scale water modeling. Due to the large spatial scales of the National Water Model, several simplifying assumptions are understandibly employed to come up with regionalized estimates of the most important parameters as determined by a sensitivity analysis. These assumptions are well characterized and the limitations of them are properly described in the assumptions. Although the overall skill improvement of the aggregated response in time and space is minor, several knowledge contributions are made in the process. An assessment of the spatial variance of the methods is properly documented and some of the major contributing factors to skill performance outside of the scope of the paper are discussed.

One of the principle questions that is a bit unclear to me is if the Land Surface Model wasn't ran in the 8 year simulation then how were the inflows and subsurface fluxes determined? Since stream flows are highly dependent on magnitude to determine their optimal, respective parameter settings, it's important to have more clarity as to where these data points were obtained. Otherwise, the limitations of the study are clearly outlined especially including the coarseness of the regionalization, lack of additional predictors in the regression, lack of spatial relationships, and lack of consideration for compound friction values. A fair survey is conducted of more robust techniques that can be employed in the future to possibly obtain better results. An assessment of the spatial variance of the performance of the methods is properly documented and some of the major contributing factors to skill performance outside of the scope of the paper are discussed. The study's conclusions are fair given the methodologies employed and the results obtained. Overall, the motivation for more work into calibrating continental scale hydrologic models is well argued for.

*We thank the reviewer for their acknowledgement of this paper's contributions despite the outlined limitations and assumptions. The reviewer is correct that inflows and subsurface fluxes were obtained from the outputs of the Land Surface Model (LSM). These fluxes were generated from a prior full model simulation over this time period using the operational settings and were used as inputs for the channel routing module simulations used in our experiment. LSM simulation settings reflect a partial calibration performed by the National Water Model team, but any further LSM simulation was beyond the scope of this analysis.*

Specific technical corrections hover around the ambiguous use of variable symbols including but limited to m, b, i, S, and w. The authors should strive to use unique notations for each variable across equations to avoid unnecessary ambiguities. More clarity can be provided when discussing the different samples of stream gages used. Please see the attached file for more technical comments.

*We recognize the confusion that may arise from repeated variable names and make several alterations to the formulas and variables to correct this and limit any confusion:*

*Original, Line 207: "$d = m \times w + b$"*

*Revised: "$d = \boldsymbol{\beta_1} \times w + \boldsymbol{\beta_0}$"*

*Original, Line 214: "$\log(n_i) = m \times \log(S_i) + b$"*

*Revised: "$\log(n_j) = \boldsymbol{\beta_1} \times \log(S_j) + \boldsymbol{\beta_0}$"*

Please also note the supplement to this comment: https://hess.copernicus.org/preprints/hess-2021-552/hess-2021-552-RC2-supplement.pdf

*The contents of this supplement PDF appear to be the original manuscript only.*